# Natural spatial pattern—When mutual socio-geo distances between cities follow Benford's law

**Katarzyna Kopczewska** ⬚ *◔, **Tomasz Kopczewski**◔

Faculty of Economic Sciences, University of Warsaw, Warsaw, Poland

◔ These authors contributed equally to this work.
* kkopczewska@wne.uw.edu.pl

## Abstract

Benford's law states that the first digits of numbers in any natural dataset appear with defined frequencies. Pioneering, we use Benford distribution to analyse the geo-location of cities and their population in the majority of countries. We use distances in three dimensions: 1D between the population values, 2D between the cities, based on geo-coordinates of location, 3D between cities' location and population, which jointly reflects separation and mass of urban locations. We get four main findings. Firstly, we empirically show that mutual 3D socio-geo distances between cities and populations in most countries conform with Benford's law, and thus the urban geo-locations have natural spatial distribution. Secondly, we show empirically that the population of cities within countries follows the composition of gamma (1,1) distributions and that 1D distance between populations also conforms to Benford's law. Thirdly, we pioneer in replicating spatial natural distribution–we discover in simulation that a mixture of three pure point-patterns: clustered, ordered and random in proportions 15:3:2 makes the 2D spatial distribution Benford-like. Complex 3D Benford-like patterns can be built upon 2D (spatial) Benford distribution and gamma (1,1) distribution of cities' sizes. This finding enables generating 2D and 3D Benford distributions, which may replicate well the urban settlement. Fourth, we use historical settlement analysis to claim that the geo-location of cities and inhabitants worldwide followed the evolutionary process, resulting in natural Benford-like spatial distribution and to justify our statistical findings. Those results are very novel. This study develops new spatial distribution to simulate natural locations. It shows that evolutionary settlement patterns resulted in the natural location of cities, and historical distortions in urbanisation, even if persistent till now, are being evolutionary corrected.

## Introduction

In recent years, Benford's law received a lot of attention. The passing decade brought half of more than 1600 papers published in the recent 140 years [1]. Benford's law assumes that the leading digits of the real-world numbers are not random and equally frequent. Conformity to

**Data Availability Statement:** Datasets are R codes for this study are publicly available. Dataset with individual geo-locations of cities and their populations and dataset with country-wise data: 1D, 2D, 3D distances, their MAD measures, Zipf

coefficients and R of Clark-Evans test, are available at Figshare (https://doi.org/10.6084/m9.figshare.c.6137520.v1). R codes to replicate the whole analysis are available at GitHub (https://github.com/kkopczewska/spatial-Benford). Contours of countries are available in rworldmap:: R software package (https://cran.r-project.org/web/packages/rworldmap/). All calculations were conducted in R software, using following packages: benford.analysis::, maps::, rworldmap::, spatstat::, GGally::, sp::. Details of running spatial analysis in R can be found in Kopczewska (2020).

**Funding:** The authors received no specific funding for this work.

**Competing interests:** The authors have declared that no competing interests exist.

this law is tested statistically using the Benford distribution, which counts the frequency of first and further digits in numbers. In natural distributions, which often follow Benford's law, digit "1" appears in the first position in 30.1%, while digit "9" as the first digit comes only in 4.6%. There are four features that make datasets potentially compatible with Benford's law: reasonable sample size–that enables least frequent values to appear, sufficient data span–to avoid all numbers starting with the same digit, right-skewed distribution–to replicate the multiplicative character of data, and no human intervention–that assures natural design [2].

Many studies confirm that natural distributions of numbers are consistent with Benford law. Its applications are mostly twofold: financial issues (frauds, audits, corruption, incomes, earnings, taxes, prices, money laundering, forensic) [3, 4]), and elections and voting [5]. Increasingly appear other Benford applications as in Covid-19, epidemiology, psychology, image processing, air quality, security and big data [1, 5], distances to galaxies [6] and complex networks [7] etc.

Until now, Benford's law was not applied in a spatial context. An interesting example of natural spatial processes is the geo-location of cities and their population. Globally, in 2016 there were ca. 1.055 cities with a population exceeding 0.5 mln [8], and ca. 4.053 cities with at least 100.000 people [9]. Urban location is one of the most persistent and natural phenomena. A rough estimation of the number of cities continuously inhabited for several centuries is around 90–100 in Europe and Asia, approximately 35 in Africa and North America and only 13 in South America. Since then, the borders of countries have shifted significantly. The current urban spatial structure is a snapshot of a long history of urbanisation. Is this socio-geographic pattern natural?

In the last hundred years, there were many concepts on how to explain the location and population of cities. Population studies mostly use the Zipf law on administrative data [10–12]–the general conclusion is that this law holds, but the results strongly depend on the methods and data applied. An interesting approach is by using night-light satellite images to address not administrative population data but a natural settlement (non-biased data), which often goes beyond the official contour of the city. Jiang et al. (2015) [13] analyses with Zipf the global urban settlement for "natural cities" and found Zipf conformity in almost 60% of countries. Decker et al. (2007) [14] in contrary for nigh-light data show lognormal distribution fits data much better than Zipf. However, Zipf law (and lognormal distribution) never considers the geographical location of the cities and does not see cities as a socio-geo network. This simplified, one-dimensional approach neglects spatial components, which seems naturally important if a mass of people concentrates in neighbouring or distant cities. There are also a few other approaches–however, they are usually based on socio-economic relations, are selective territorially, consider major cities only, refer to a relatively short time span and are not anchored in history. Dziecielski et al. (2021) [15] offer a good review of those studies, from Christaller's Central Place Theory and New Economic Geography (NEG), through Tinbergen-Bos and Tellier models to Systemic Economic Geography (SEG) focusing on neo-Newtonian approach to see "spheres of influence" and gravity forces of 14 European urban agglomerations by using data on accessibility, cultural interactions, economy, environment, liveability and R&D. Ioannides and Overman (2004) [16] try (and fail) to explain urban system with city location and their socio-economic interactions. Most modern view of geographic space as a living structure explains human activity (like Tweeting) in an interesting way with a topological representation of the built environment and big data using the case of millions of street nodes of the United Kingdom [17].

Our study approaches this old problem of urban geography very differently. We treat the geographic system of cities as a complex non-linear three-dimensional network, which is not random or independent but reveals high spatial heterogeneity. We search for hidden order

which is to be explained with Benford law. We consider the socio-geographic three-dimensional (3D) space (localisation and population) and its components: two-dimensional geographic distance (2D) and population one-dimensional distance (1D). As Benford law holds in natural datasets, we claim that spatial phenomena like urbanisation, when created naturally, are Benford-like. In contrast, spatial processes driven by the other (e.g. political) rules of ordering fail this conformity. In the natural spatial pattern of urban settlement, spatial separation and mass of cities are not random but constitute a complex three-dimensional network conforming with the ordinary course of nature.

The study's design is not typical but builds a complete story. The first section describes the statistical methodological approach applied in this research, including Benford law and MAD statistic, Euclidean distance, Clark-Evans test for point patterns and Zipf law. In the second section, we analyse empirical urban data worldwide in the search for Benford conformity. We also compare the conformity of population data with Benford and Zipf law. In the third section, we show how to replicate in spatial statistical simulations the socio-geographical Benford-like distribution using mixtures of generated point patterns and populations. This study develops new spatial distribution to simulate natural locations and shows that $n$-dimensional space (here 3D socio-geo space) can be reduced to a one-dimensional variable of distances. In the fourth section, we link our statistical results with a historical overview of a few hundred years of settlement. It shows that evolutionary settlement patterns resulted in cities' natural location, and historical distortions in urbanisation, even if persistent until now, are being evolutionary corrected. It also confirms the findings from statistical simulation.

## Methodological approach to study urban locations and their populations for Benford's law conformity

This section presents statistical tools which will be used in further analysis: multidimensional Euclidean distance, Benford distribution, MAD (*Mean Absolute Deviation*) conformity, Clark-Evans test of spatial randomness in point pattern and Zipf coefficient. The study is designed for data that include geo-locations (*xy*) of cities worldwide, the population of those cities (*z*) and information on the country and continent to which the city belongs. Mutual relations between cities–their separation in space and mass can be captured by calculating distances between them. We distinguish three kinds of distances:

- 1D (one-dimensional) distance–between population size of cities

- 2D (two-dimensional) distance–between locations of cities, based on their geo-coordinates

- 3D (three-dimensional) distance–between locations of cities and their size jointly.

Most intuitive is 2D distance, which expresses how far (e.g. in km) is from city *i* to city *j*. However, it does not include information on how many people live in a given location. Thus, 3D distance jointly considers separation and mass–for example, two cities distanced by 100 km and each inhabited by 1 mln people are "closer" to each other than if their populations were 0.1 mln and 1.9 mln.

One of the most fundamental and flexible metrics is Euclidean distance. Most comprehensive, 3D distance is expressed as:

$$dist_{ij,3D} = \sqrt[2]{(x_i - x_j)^2 + (y_i - y_j)^2 + (z_i - z_j)^2} \tag{1}$$

where *i* and *j* are different cities within a given country, *x* and *y* are the longitude and latitude of the city, and *z* is the city's population. 1D and 2D distances can be obtained from the formula for 3D distance by assuming redundant dimensions as 0. 1D distance is, in fact, a

difference between population sizes in both cities. 2D is the shortest path between two locations and does not consider natural borders or roads.

The 1D, 2D and 3D distances will be tested for Benford's law conformity. In a set of numbers, if the leading string of numbers $d$ (for first digit $d \in \{1,\ldots, 9\}$ or first two digits $d \in \{10,\ldots, 99\}$) appears with probability given with (2), the numbers satisfy Benford's law and are naturally distributed:

$$P(d) = log_{10}(d + 1) - log_{10}(d) = log_{10}\left(\frac{d + 1}{d}\right) = log_{10}\left(1 + \frac{1}{d}\right) \tag{2}$$

Benford's law is independent of metric system and scale [5, 18]. One can test conformity with Benford in many ways [7]. However, recent discoveries [19] indicate that many of the tests are too sensitive, giving false positives too frequently. MAD (*Mean Absolute Deviation*) Conformity [20, 21] works the best among available testing methods. Its correction, Excess MAD, can limit the number of false positives (*too often not conforming*) in a small sample (min. 1'000 obs.) [22]. Two-digits MAD, the most often applied test for Benford's law conformity, is expressed as:

$$MAD = \frac{1}{90} \sum_{k=10}^{99} \frac{|Obs_k - Exp_k|}{N} \tag{3}$$

while

$$Excess\ MAD = MAD - \frac{1}{\sqrt{158.8 \cdot N}} \tag{4}$$

where N is the sample size, while $Obs_k$ and $Exp_k$ are observed and expected quantities of two-digit numbers, comprising of given two-digit string. MAD counts the observed frequencies of the first two digits $Obs_k$ and compares with the expected frequency $Exp_k$. In general, one needs at least 10'000 observations for very good conformity. Values of MAD between 0.000–0.0012 are interpreted as *close conformity*, between 0.0012–0.0018 as *acceptable conformity*, between 0.0018–0.0022 as *marginally acceptable conformity* and above 0.0022 as a *nonconformity* [21]. In this paper, all three conformity levels (close, acceptable and marginally acceptable) will be treated as conformity with Benford's law.

As the locations of cities create the spatial pattern, it will be tested within countries against CSR (Complete Spatial Randomness). We consider three major types of spatial patterns: random (*Poisson*), ordered (organised, uniform) and clustered (agglomerated), which find justification in the history of urbanisation. We have applied the **Clark-Evans test** [23] (Clark & Evans, 1954) for Complete Spatial Randomness (CSR) of spatial distribution, which is considered stable and reliable [24, 25]. The Clark-Evans test shows the direction of non-CSR tendency, expressed as an alternative to the null hypothesis–it indicates if the non-random spatial pattern is clustered or ordered. It uses the average nearest-neighbour distance, while the statistic—aggregation R index—is the ratio of those distances in empirical and theoretical (random) point patterns. It is expressed as:

$$R = \frac{\bar{r}_{empirical\ locations}}{\bar{r}_{random\ locations}} = \frac{\sum r}{n} : \frac{1}{2\sqrt{n/s}} \tag{5}$$

where $r$ is the distance from a given point to the nearest neighbour, $\bar{r}$ is the average $r$, $n$ is the total number of points, and $S$ is the total analysed area. In clustered patterns, the average distances are shorter than in random spatial distributions, thus ratio R<1, while in ordered

pattern oppositely, therefore R>1. A variance of R is $var(R) = 0.2732/n$ and can be used in t-test ($t_R = (R - 1)/\sqrt{var(R)}$) for significance [26].

Benford law, together with Zipf law, belong to a power law (e.g. [18]). Even if they behave similarly being heavy-tailed distributions [27], they keep different properties [18]. Zipf law is traditionally applied to analyse the size of cities within a given country. Zipf's coefficient in the Pareto form is usually calculated as a regression coefficient (slope) from the linear model, in which the dependent variable $y$ is the log of ranks of cities (the biggest city is no 1, the smallest city is the last number), while explanatory variable $x$ is the log of the population of cities. The Lotka form assumes opposite relation, that log of population (y) is explained with long of rank (x) [11]. Regression coefficients around -1 evidence intra-country Zipf's law conformity. Many studies confirm that urban populations follow the Zipf law (e.g. [13]), but also meta-analysed of Zipf law studies show that there is no single Zipf law [11] and Zipf results differ between disciplines and the focus of study [12].

Using Zipf in this study has two functions: first, it builds a well-understood reference to other studies dealing with urban populations; second, it is to show the difference between Zipf and Benford. The fundamental difference is that Zipf never refers to precise geo-location of cities of which population is being studied–Zipf is locationally invariant—even if cities with their populations would be relocated, Zipf value will stay the same. Oppositely, Benford in our approach with 3D distance, has important spatial features. To visualise it better, analysis of Zipf is on a vector of population, while Benford's on a complex three-dimensional network of cities and inhabitants, where the reshuffling of urban locations or their populations or both matters for the outcome. The analysis presented below is twofold. In empirical section, we show the worldwide Benford conformity for 3D distances, and we run statistical analysis exploring the relations among all mentioned measures: MAD for 1D, 2D and 3D distances, Zipf coefficient, R of Clark-Evans test, and we check the conformity with Benford's law against a number of cities within the country, with regard to the continent, size and shape of the country, and to the point pattern of cities observed within the country. In simulation section, we run a simulation to replicate spatial (2D) and soc-geo (3D) Benford distribution. Therefore, we simulate different point patterns and their mixtures in changing proportions and add population values to generated locations. We follow a systematic approach in selecting the parameters of consecutive iterations.

## Empirical study on the conformity of urban system with Benford's law

In an empirical study we have used the dataset including originally 43'645 cities from 239 countries with their geo-location (*lon*, *lat*) and number of inhabitants within administrative borders. Dataset was limited by 1'617 cities and 70 countries by eliminating countries with few cities and the very small island countries where the urban location is pre-determined by the island's shape. We conducted analysis on $n$ = 42'028 cities in $i$ = *169* countries. The number of cities per country varies from 11 to 1'000.

Dataset was collected from Gazetteer in 2006 and made available as *world.cities* in maps:: R package. Data including calculations presented in this paper are available at Figshare [28]. All calculations were conducted in R software, using the following packages: *benford.analysis*::, *maps*::, *rworldmap*:: and *spatstat*::. Details of running spatial analysis in R can be found in [29].

### Conformity of 3D distances with Benford's law

Within each country $i$, we have calculated the matrix of $\left(\frac{n_i^2 - n_i}{2}\right)$ mutual 3D (*xyz*) socio-geo

Euclidean distances between all available $n_i$ locations (cities in given country) and their

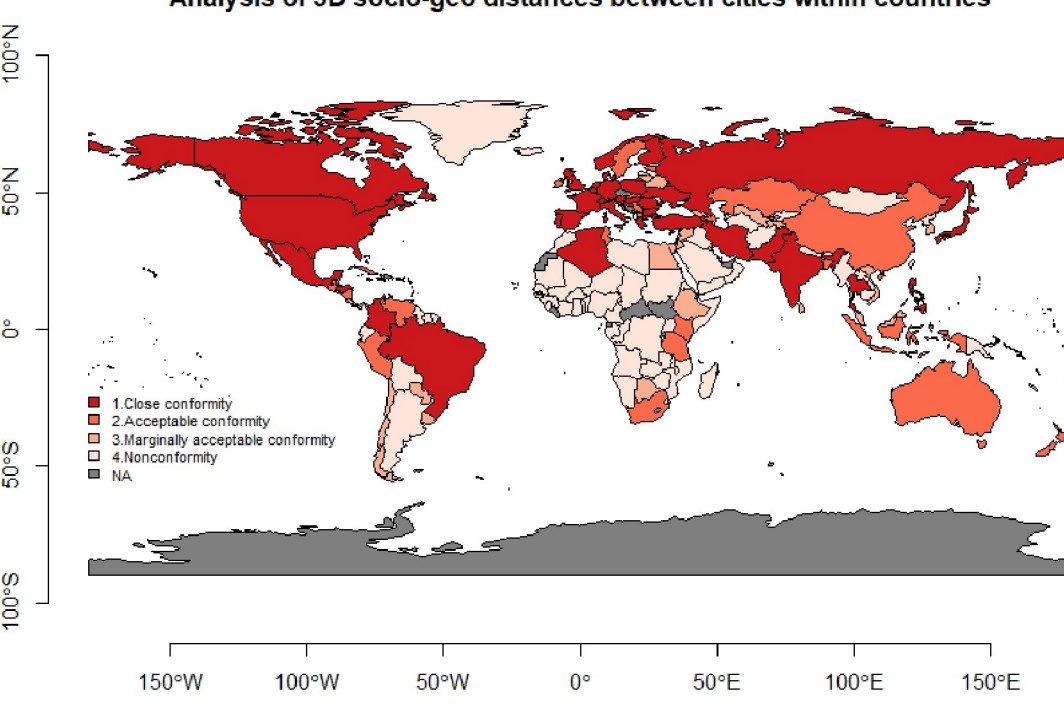

**Fig 1. Conformity to Benford distribution—analysis of 3D mutual socio-geographic distances between cities within countries.** Source: Own study with the use of benford.analysis::, maps::, rworldmap:: R packages.

population. Due to symmetric distances between given two cities, a number of mutual distances $n_i^2$ can be limited by half; one also eliminates zero distances between each location with itself. Country by country, we tested if the set of the socio-geo distances conforms with Benford's distribution. It was accompanied by the same analysis of 2D geo-distances ($xy$) and 1D population ($z$) distances. Countries with a maximum number of cities $n_i = 1000$ (e.g Greece, France) have 499'500 mutual connections to analyse, middle-size countries with $n_i = 100$ cities have 4'950 links, while a minimum number of links (55) appears in the case of a country with $n_i = 11$ cities (e.g. Bahrain). In total, we analyse more than 13 mln connections.

Results of this analysis (Fig 1, Table 1) provide evidence that in most countries (90 out of 169), the 3D socio-geo distances between cities conform with Benford's law. However, continents differ– 83% of countries in Europe, 58% in Asia and North America, 54% in South America, and only 23% in Africa conform with Benford's law. Australian countries are mostly organised due to the island's shape, which limits the choice of urban location. We discover that despite the different location patterns of diversified cities among countries (due to the natural conditions, the shape of the country, population density, history etc.) the urban organisation worldwide follows a natural spatial pattern. What does it mean? For ages, people, in their decisions where to locate, behaved naturally, thus spatial separation and mass of cities are not random but constitute a complex three-dimensional network conforming with the ordinary course of nature.

**Table 1. Statistics by continents for conformity with Benford in distributions of 3D socio-geo mutual distances.**

| Continents | Close conformity | Acceptable conformity | Marginally acceptable conformity | Nonconformity | Total |
|---|---|---|---|---|---|
| Europe | 23 | 9 | 2 | 7 | **41** |
| Asia | 8 | 7 | 10 | 19 | **44** |
| North America | 4 | 3 | 2 | 7 | **16** |
| South America | 2 | 3 | 3 | 5 | **13** |
| Africa | 1 | 6 | 4 | 37 | **48** |
| Australia | 0 | 3 | 0 | 4 | **7** |
| **Total** | **38** | **31** | **21** | **79** | **169** |

Note: Continents in rows, MAD conformity criteria in columns, counts of countries in which mutual 3D socio-geo distances between cities conform with Benford's law
Source: Own study with the use of benford.analysis:: R package.

## Determinants of Benford's law conformity

Below we try to explain the conformity of 3D socio-geo distances with Benford distribution with diverse factors. We analyse sample size, nature of data, the behaviour of spatial point pattern, Zipf's law, mutual relations between different factors, and size or shape of the country.

## Sample size

Benford's law conformity is big-sample law. For a few observations, Benford conformity is rarely discovered [30], while stable outcomes require at least *n>1.000*, and preferably *n>10.000* observations [31]. This study checks mutual distances in big and small countries. We discover that Benford conformity (close, acceptable, and marginally acceptable) requires at least $n_i = 100$ cities within the country, which is equivalent to $\left(\frac{n_i^2}{2} - \frac{n_i}{2}\right) = 4950$ mutual distances and grows with the number of sites (Fig 2). The majority (86 of 99) of countries with at least 100 locations conform with Benford distribution (closely, acceptable or marginally). The

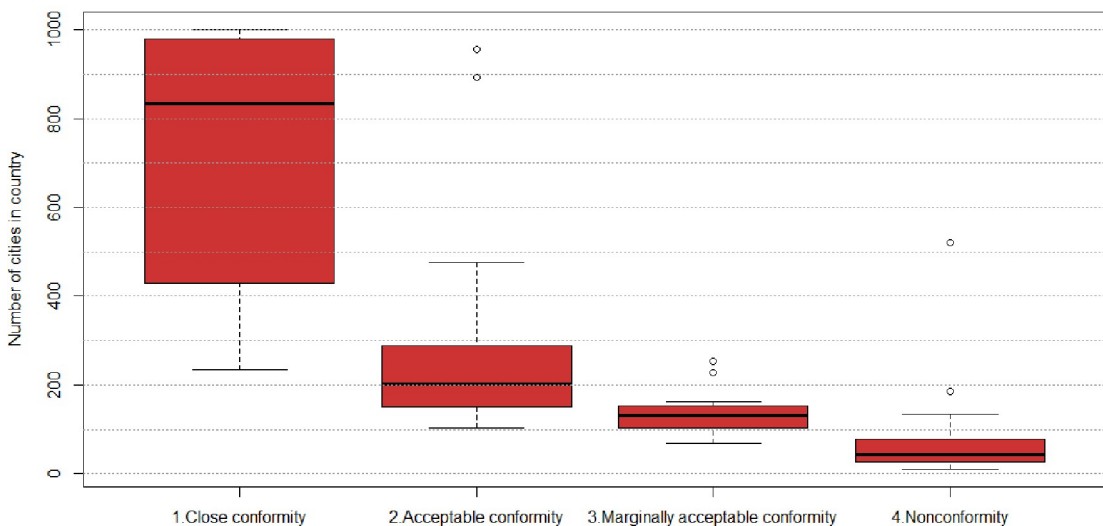

**Fig 2. Relation between conformity to Benford distribution of 3D socio-geographic distance and number of cities in a country.** Source: Own study with the use of benford.analysis:: R package.

majority of small countries(66 out of 70) with less than 100 urban locations reported, were classified as non-conforming with Benford (Fig 2).

The threshold of 100 cities to distinguish between big and small countries was taken from statistical outcomes (as the bottom part of the box in marginally acceptable conformity). However, it raises an important question of whether globalisation processes, trading, migrations and other interactions impact the analysis. We adopted the institutional perspective, which assumes that resettlement of average inhabitants is much easier within the country than between the countries due to language barriers, formalities, different institutional systems, permissions, certifications, schooling systems, etc. This approach enhances the role of national borders. This spirit of analysis can be confirmed by the study of the US urban system by [16], who claim that interactions between the cities cannot explain the size of cities (and thus mobility between them). However, this is an issue for further investigations.

**Spatial point-patterns.**   we studied the spatial distributions of cities within the countries. We used the Clark-Evans test, which may detect if the point pattern is random (R≈1) or clustered (R<1) or ordered (equally distributed) (R>1). There were just a few studies using this approach–[16] found that US cities (small sample up to 332 locations) were clustered in ca.1900 while later were randomly distributed; [32] found that gridded population in African countries is very diversified in values of Clark-Evans R, but mainly point patterns are clustered. However, there is no evidence in the literature on 3D distances and links to Benford law.

We find that geo-location of cities (2D, point-pattern) impact the 3D Benford conformity. Even if 2D mutual geo-distances ($dist_{xy}$) were Benford-like only in 7 of 169 countries, there exists a very strong positive correlation (ca.0.74) between MAD values of 2D-geo and 3D-soc-geo distances (Fig 6), so Benford's law conformity in 2D and 3D is linked.

We discovered that 3D socio-geo Benford conformity appeared at different spatial distributions. The majority (53 out of 90 = 59%) of Benford-confirming 3D distances are based on the clustered (agglomerated) pattern. Simultaneously, two other patterns are also not an exception–ordered appeared in 23% (21/90), and random locations in 18% (16/90) of countries (Table 2, Fig 3). This proves that the Benford pattern is not another pure type of spatial pattern. Instead, we claim that Benford's law conformity appears in mixed spatial distributions, which link random, clustered and regular distributions in some proportions. However, the more regular the pattern, the higher the 3D distance MAD (corr = 0.4) and the lower the chance for Benford conformity. We also find that in countries with less than 100 cities Clark-Evans test detected a predominantly random pattern (43%, 30/70) and ordered pattern (37%, 26/70), while the agglomerated pattern was revealed only in 20% of cases (14/70). As Benford conformity mostly appears in agglomerated patterns, this justifies why Benford conformity was rarely detected in small countries. What remains unpuzzled here is if countries with less than 100 cities indeed do not have clustered patterns, or is it the statistical effect that agglomeration is hard to detect in a small sample.

**Population distribution and Zipf's law conformity.**   for population data, we checked Zipf's law conformity. We did not restrict data within the countries by setting any threshold for city size. All cities available in the database were used in calculations to keep coherence with the Benford analysis. As we focus on complex city network analysis, we were interested in a possibly comprehensive picture of the urban system. We evidence that only in a few countries (23 out of 169, 14%) Zipf coefficient is around -1 (±0.05, from -0.95 to -1.05), while in a majority (123 out of 169, 73%) the Pareto form Zipf coefficient was between (-0.37, -0.94), and only in few countries (23/169, 14%) its absolute value exceeded -1 (-1.056, -1.411) (Fig 4). This marginal share was already confirmed in the literature by [33], who analysed what number of cities in a subsample conforms with Zipf–their results show that in the case of the U.S. the Zipf conformity requires between 140 and 205 cities, and for more or less cities it disperses from 1

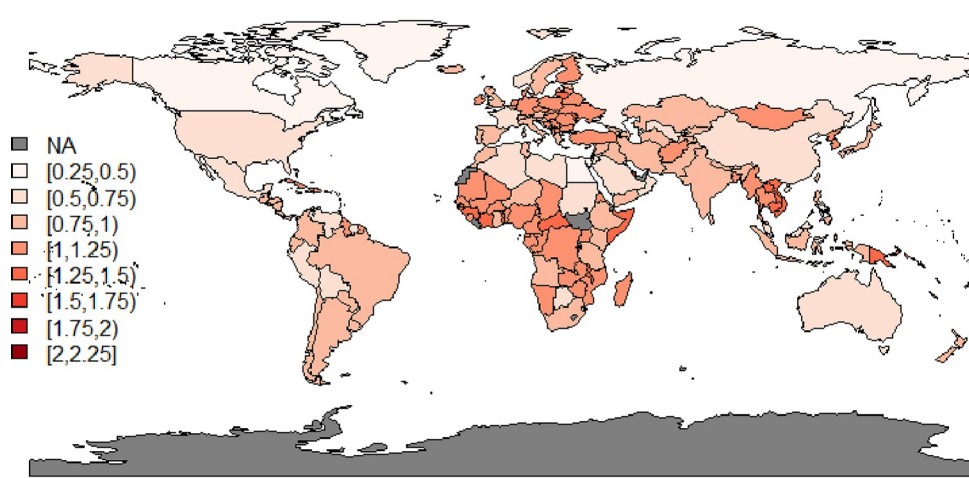

R<1 represents agglomeration, R>1 represents ordered distribution, R=1 is for CSR

**Fig 3. Map of R-value from Clark-Evans test.** Source: Own study with the use of spatstat::, maps::, rworldmap:: R packages.

significantly. Cristelli et al. (2012) [10] also confirm that Zipf holds conditionally in selected subsamples only. Countries with Zipf-like population distribution are diversified: the number of cities varies from 27 to 985, the coefficient of variation of city size within the country is between 0.84 and 7.4, Clark-Evans R is between 0.42 and 1.25, and countries are on all continents.

**Table 2. Relation between Benford and spatial 2D distributions.**

| MAD Conformity for Benford test | Agglomerated point-pattern | | Ordered point pattern | | Random point-pattern | | Total | |
|---|---|---|---|---|---|---|---|---|
| | Whole sample | Countries with 100 + cities | Whole sample | Countries with 100 + cities | Whole sample | Countries with 100 + cities | Whole sample | Countries with 100 + cities |
| Close conformity | 23 | 23 | 12 | 12 | 3 | 3 | 38 | 38 |
| Acceptable conformity | 18 | 18 | 6 | 6 | 7 | 7 | 31 | 31 |
| Marginally acceptable conformity | 12 | 10 | 3 | 2 | 6 | 5 | 21 | 17 |
| Nonconformity | 17 | 5 | 28 | 3 | 34 | 5 | 79 | 13 |
| Total | 70 | 56 | 49 | 23 | 50 | 20 | 169 | 99 |

Note: MAD conformity criteria in columns, pure point-patterns in columns, in table counts of countries in which spatial distribution of cities conform with pure point-pattern. Data were separated into the whole sample (with big and small countries) and big countries (with 100+ cities). The threshold for significance of Clark-Evans test was p-value = 0.05

Source: Own study with the use of benford.analysis:: R package.

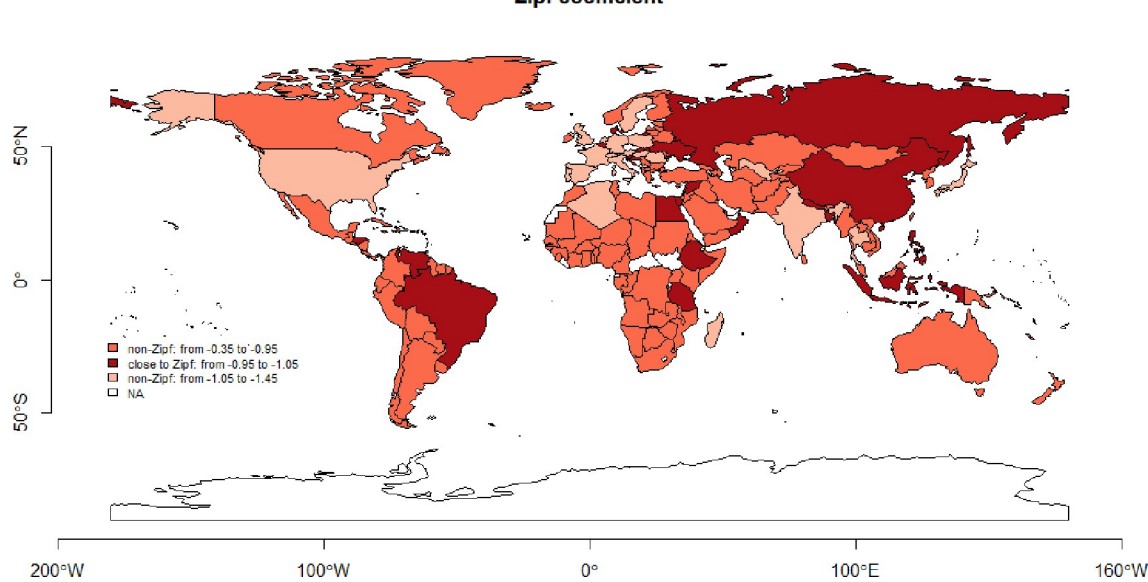

**Fig 4. Zipf's law conformity within countries.** Source: Own study with the use of maps::, rworldmap:: R packages.

The correlation between Zipf's coefficient and 3D MAD is average (0.41)—the higher Zipf's coefficient (the more similar / less diversified the cities within the country), the higher MAD (the weaker Benford conformity). However, this relation is not strong, and the information included in Zipf and Benford differs. This is in line with the statistical properties of Benford distribution. In general, statistical distributions which are around the normal or regular pattern and tend to follow symmetry almost never follow Benford [34]. Much more Benford-fitting are survival-like distributions, where the frequency of occurrence is an inverse function of the object's size–when smaller objects appear more often than the bigger one and objects are more diversified.

This finding is in line with the observation of [33], who show that in the case of datasets for China, the Zipf coefficient equal 1 is rarely achieved. They explain this with a long-term policy of balancing the size of cities by limiting the growth of big cities, reasonably developing middle-size cities and strongly supporting the growth of small cities. However, in light of the results of this study, policy based on a non-spatial vector of population data lacks the complex network features inherited in 3D Benford approach. Thus, deeper conclusions from this part of the study are not justified.

For simulation needs, we checked the consistency of population empirical distribution (within countries) with theoretical statistical distributions. We found out that those distributions are highly inhomogeneous, and there are three local density extremes mostly. In the statistical analysis of standardised data, we found that it is a **composition of three *gamma* (1,1) distributions**

Division into three subgroups of 90% observations (percentiles 0%-90%), 9% of observations (percentiles 90%-99%) and 1% of observations (percentiles 99%-100%) and standardisation within those subsamples allows for getting approximately similar group distributions *gamma* (1,1) (Fig 5).

**Mutual relations.**    even if in the original dataset a number of variables is limited, various computations which expanded the dataset allow for deeper statistical analysis of mutual relations between diverse measures. In Fig 6, we analyse relations between a) Zipf's law (*Zipf*) (its

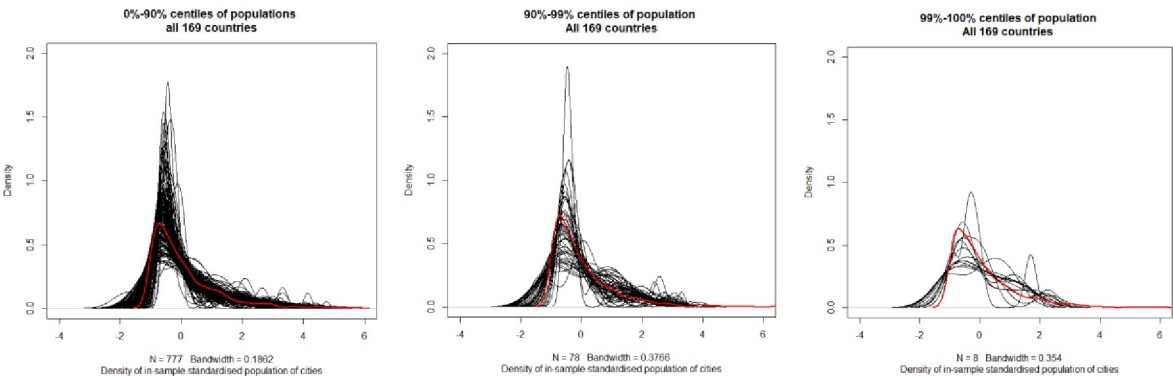

**Fig 5.** Consistency of empirical population distribution (standardised in subgroups) with gamma (1,1) distributions: a) 0–90% centiles, b) 90–99% centiles, c) 99–100% centile. Note: in countries with less cities, top centiles have just few observations, which are not plotted as distribution. Source: Own study with the use of R software.

value around -1 confirms Zipf's law, while -1>coeff>0 suggest over-representation of big cities), b) R in Clark-Evans test (*RofClarkEvans*) (it tests the type of point-pattern of geo-located data: R<1 clustered, R = 1 random and R>1 ordered locations), c) MAD statistics for Benford

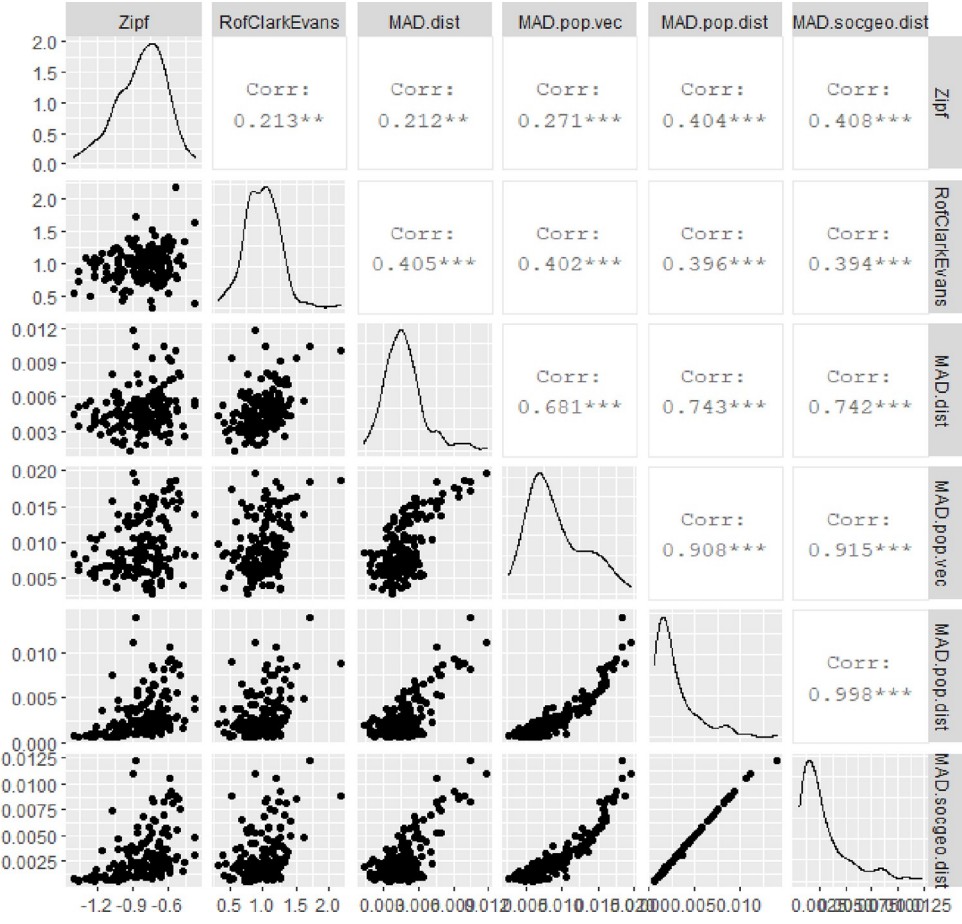

**Fig 6. Relations in empirical data.** Source: Own study with the use of GGally:: R package.

conformity for 2D distances between spatial points (*MAD.dist*) (the higher value, the less Benford-like spatial pattern), d) MAD statistics for Benford conformity for a vector of population values (*MAD.pop.vec*), e) MAD statistics for Benford conformity for 1D distances between population values (*MAD.pop.dist*), f) MAD statistics for Benford conformity for 3D soc-geo distances between locations and population (*MAD.socgeo.dist*).

This figure shows somehow unexpected results. Firstly, even if Benford conformity for a vector of urban population (*MAD.pop.vec*) did not appear in any country, and for a matrix of distances between urban populations (*MAD.pop.dist*) it occurred in 90 countries (53%), their MADs correlation is high (0.91). It proves soft relation, which is sensitive to MADs thresholds. Distributions of populations and distribution of mutual distances between populations are linked. Secondly, one can observe a very strong positive correlation (ca.0.74) between MAD values of 2D-geo and 3D-soc-geo distances, even if 2D mutual geo-distances (*MAD.dist*) are rarely Benford-like (only 7 of 169 countries), while 3D-soc-geo distances (*MAD.socgeo.dist*) are often Benford-like (70 of 169 countries). Thirdly, Zipf's law is only slightly connected to Benford's law. Zipf for population does not bring the same information as Benford for population (*corr = 0.27* with MAD on population vector and *corr = 0.4* with MAD on population distance), link remains moderate when location additionally considered (*corr = 0.41* with MAD on 3D soc-geo distance). However, one can understand that over-representation of big cities (high Zipf's coefficient) weakens Benford conformity (high MAD). Zipf is also loosely linked to spatial patterns of urban locations (*RofClarkEvans*) (*corr = 0.21*) (in fact Zipf is a-spatial metric).

**Size or shape of the country.** Benford's law conformity of 3D mutual soc-geo distances between cities does not depend on the size or shape of the country (Fig 7). Our study evidences that different spatial distributions (given with R from the Clark-Evans test) and with urban locations on whole or part of territory, can conform with Benford. This applies also to size–small and huge countries, with regular or narrow shape, follow Benford distribution. We show for comparison four examples: a) USA (spatial range of 50˚, R = 0.49, Benford Close Conformity); b) China (spatial range of 50˚, R = 0.729, Benford Acceptable Conformity); c) Poland (spatial range of 10˚, R = 1.225, Benford Close Conformity); d) Italy (spatial range of 10˚, R = 0.875, Benford Close Conformity). The exemplary countries are very diverse, but all conform with Benford law.

## Discussion of results

All those analyses show that both factors, allocation of the population as well as the location of cities, matter for natural distributions, and Benford's conformity is an output of very special complex settings. This confirms the hypothesis that cities in the three-dimensional setting of location and population constitute a complex non-linear, non-random network. We claim that we discover hidden order with Benford law, which explains the observed spatial heterogeneity. This interdependency of human location decisions in space is part of the Paretian way of thinking [35] and evidences that the assumption of the Gaussian linear world is too simplistic. This finding has a critical implication for the simulation study. As indicated by [35], spatial heterogeneity, manifesting in uneven distributions, should be perceived as a mix of concentrations in a human-made built environment. In next section we show that, indeed, a mixture of point patterns and distributions may replicate this natural urban geographic system.

One should underline that the presented analysis is much more statistical than economic or political. We analyse all world countries, which naturally are at very different stages of development, facing diversified problems, running various policies, and being not uniform culturally or politically. All those factors are neglected, and only the city's location and population are

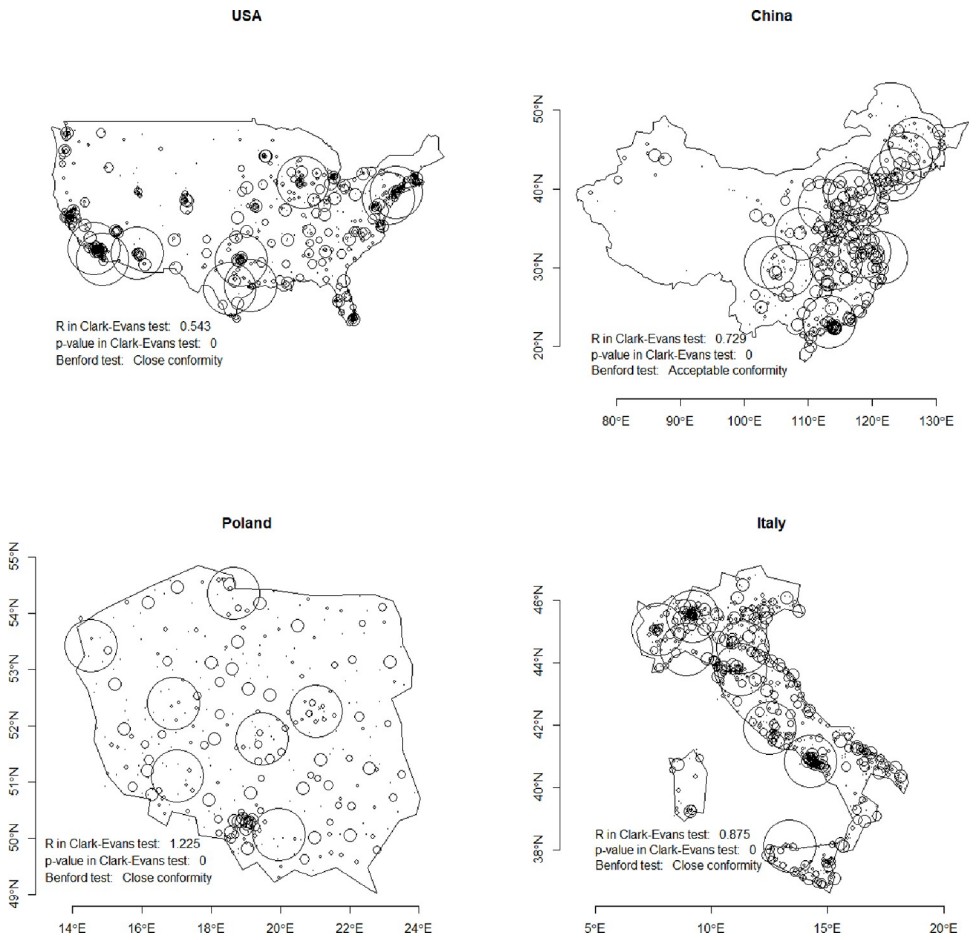

**Fig 7.** Locations of cities in selected countries, where for 3D socio-geo distance the conformity to Benford's law was achieved: a) USA, b) China, c) Poland, d) Italy. Source: Own study with the use of benford.analysis::, spatstat::, maps::, rworldmap:: R packages.

considered. This, of course, limits the range of study but allows for focusing on statistical phenomena and long-term evolutionary processes.

## Methods of replicating natural spatial distribution

Discovering the rules that make 3D mutual socio-geo distances conform with Benford's law will enable simulating the natural spatial distribution–location and population of cities. This issue was not solved until now. The main challenge lies in mutual relations, inter-connectedness, and data complexity. Firstly, unlike in simple data where one can cut off the not-fitting observation, change of single point in complex network impacts the $n_i-1$ distances. Secondly, except for generating Benford-like 2D spatial pattern ($xy$), and/or Benford-like values ($z$), there is an issue of assigning values to locations, which increases the number of possible combinations.

We simulated pure point patterns (clustered, regular, random) and their mixtures as well as population values. We tested random and ordered allocations of values ($z$) to places ($x,y$). Details of the simulation are described below. R codes for simulation are available at Github (https://github.com/kkopczewska).

## Distribution and simulation of population

Based on approximated *gamma (1,1)* distribution for population, we have generated three theoretical distributions in proportions of percentiles, as mentioned previously—by dividing urban populations within the country into three subgroups: of 90% observations (percentiles 0%-90%), 9% of observations (percentiles 90%-99%) and 1% of observations (percentiles 99%-100%)–which are standardised data. To obtain nominal (not standardised) values, we un-normalised data ($x'_i = x_i \cdot \sigma + \mu$). We assumed parameters of three subsamples close to real datasets $(\mu, \sigma, n) = \{(1,1,90\%), (8,5,9\%), (36,30,1\%)\}$. For Benford conformity, one needs to subtract one from all theoretical gamma values ($x'_i = (x_i - 1) \cdot \sigma + \mu$). Combined gamma distributions with parameters as mentioned for $x'_i = (x_i - 1) \cdot \sigma + \mu$ give 100% Benford conformity (45% close and 55% acceptable conformity when data shifted by one), while without a shift by 1, for $x'_i = (x_i) \cdot \sigma + \mu$ give 30% marginal Benford conformity and 70% nonconformity.

## Simulation of 2D point-patterns

Below we show the simulation design to discover the natural point pattern which conforms with Benford in two steps: a) individual analysis of pure point patterns (clustered, regular, random), b) analysis of different mixtures of spatial distributions.

The first point is to analyse the point locations and Benford distributions of mutual 2D distances between geo-located points. We have considered three pure point patterns: regular (organised), clustered (agglomerated), or random (homogenous Poisson, CSR–Compete Spatial Randomness) (Fig 8A and 8B). We have generated $n_i = 1000$ observations for each pattern within the square box (*bounding box*). The ordered pattern (*red*) was generated as nodes of grid-like locations. Random spatial distribution (*yellow*) was a typical Poisson point pattern. Locations in clustered distribution (*blue*) were simulated as a two-dimensional normal distribution, with $mean_1 = 0.5 \cdot range$ or $mean_2 = 0.25 \cdot range$ and $sd = 0.15 \cdot range$, where $range = x_{max} - x_{min}$ of the bounding box, $mean_1$ is in the centred-core pattern (Fig 8A), and $mean_2$ is in the shifted-core pattern (Fig 8B).

The density of mutual distances tends towards symmetry in random and ordered point patterns and behaves like survival function in clustered spatial distribution (Fig 8C). Those densities follow statistical distributions; however, the result–what distribution it follows–may depend on distance metric as well as the shape of the bounding box of the point pattern [36]. This analysis was skipped here. In analysed point patterns, the Clark-Evans test reacts appropriately in all cases, giving $R \approx 1$ for random pattern, $R \approx 2$ for regular pattern and $R \approx 0.6$ for a clustered pattern (all at *p-value*<0.01). However, neither of the mutual-distances matrices of pure patterns conforms with Benford (Fig 8D)–theoretical Benford distributions (*red*) are far for the empirical ones (*black*). The clue is that clustered point-pattern, because of its survival-like shape of density, has the biggest chance to be the driver of natural Benford-like spatial distribution [2], while the other two patterns are to supplement it.

The second point of analysis, to check the mixtures of point patterns, is based on expectations from cities' empirical studies. We used simulation to detect if there is any combination of proportions of three basic spatial distributions, which give mutual distances conforming with Benford's law. We ran thousands of drawings from a uniform distribution [0,1] to obtain frequencies of two point-patterns $s_1$ and $s_2$, while the share of the third one $s_3$ was calculated as $1 - s_1 - s_2$. We used 4.000 iterations with all positive $s_i$ only. For each selected iteration, we drew in total $n_i = 1.000$ locations (thus *499'500* mutual distances), where proportions of distributions used were given by $s_i$. In the case of a random and ordered pattern, the location of points is very even on the plane. Still, in clustered distribution, the core may be centred (Fig 8A) or shifted towards the border (Fig 8B). This is important for the output. A shifted core implies a

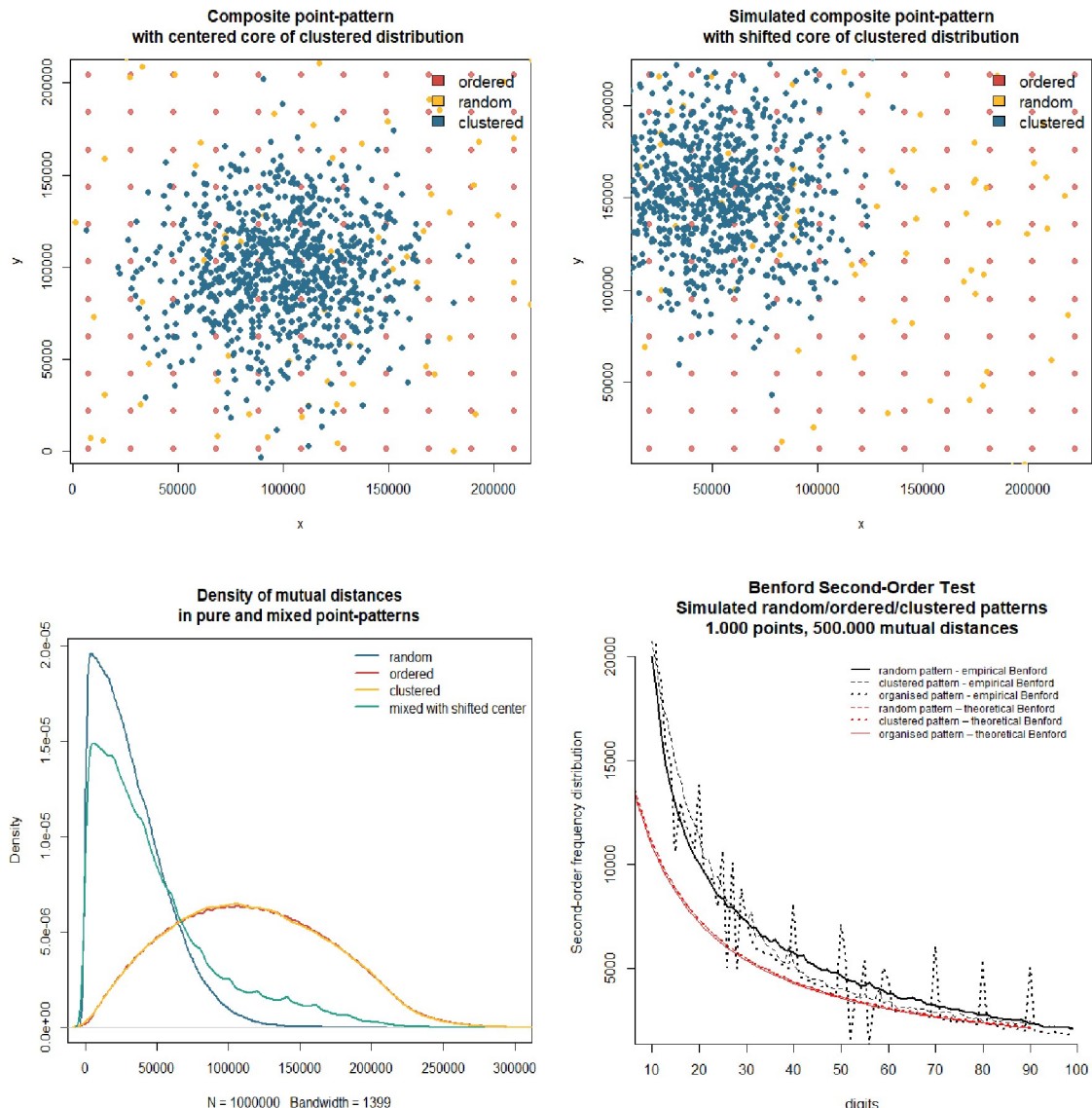

**Fig 8. Spatial and Benford distributions in pure theoretical point-patterns.** a) simulated pure point patterns—geo-location of points with centred-core cluster, b) simulated pure point patterns—geo-location of points with shifted-core cluster, c) density functions of mutual distances in pure point patterns, d) Benford's conformity of mutual distances in pure point patterns. Source: Own study with the use of sp:: and benford.analysis:: R packages.

more skewed distribution of mutual distances, which is in favour of Benford's law conformity. We tested two scenarios and obtained very robust results.

A mixed point pattern with a shifted core of the cluster (Fig 8B), has a great potential to reveal Benford conformity of mutual distances when the share of clustered distribution exceeds 50% (Fig 9). In our study, 87% of simulations with at least 50% of points coming from agglomerated point-pattern conformed with Benford's law (25% Close conformity, 43% Acceptable conformity, and 19% Marginally acceptable conformity). The relation is strictly visible at Fig 9 –the higher the share of clustered pattern, the stronger Benford's law conformity. However, the mixture is a must–the pure clustered pattern does not conform with Benford (Fig 8D). For close conformity, one needs a clustered pattern in ca. 75–80%, and 10% up

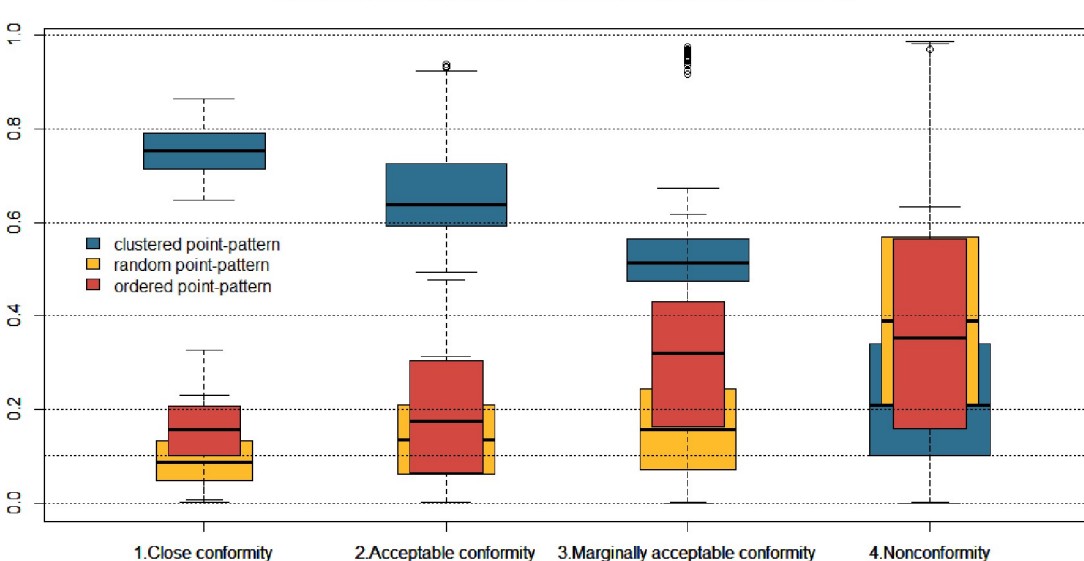

**Fig 9. Simulated mixture of point patterns to test Benford.** Source: Own study with the use of R software.

to 15% of random and organised patterns (Fig 9). Lowering the share of the clustered pattern below 50% mostly causes mutual distances to stop to conform with Benford. This proves that natural spatial distribution is a mixture of three theoretical distributions, with the significant majority of clustered points. Equalised proportions of point patterns in the majority of cases give Benford-non-conforming pattern. To sum up, we discover that proportions 15:3:2 of pure spatial distributions (ca. 75% clustered, 15% regular and 10% random) make the mutual distances between geo-located points (2D) conforming to Benford's law (Fig 9).

## Simulation of complex 3D Benford conformity

In the simulation of 3D socio-geo distances, there are at least three aspects important: firstly, in a spatial pattern, conformity or nonconformity with Benford; secondly in population data, conformity or nonconformity with Benford; thirdly, allocation of population values to locations, which links 1D and 2D distributions and impacts 3D links. On this basis, we summarise below a few simulation scenarios. Components of scenarios are as follows:

1D –distances between population–high Benford conformity for ($x_{i.gamma}$-1), low Benford nonconformity for ($x_{i.gamma}$) and nonconformity ($x_{i.gamma}$+1). Benford-like distribution of population, consistent with real data, can be generated as a composition of three *gamma*(1,1)-1 distributions, rescaled with parameters ($\mu,\sigma,n$) = {(1,1,90%), (8,5,9%), (36,30,1%)} (small, intermediate and high values) as described earlier.

2D –distances between locations–high Benford conformity for well-mixed skewed point-pattern with one non-centred cluster (as described in previous section), low (marginally acceptable) Benford conformity for well-mixed skewed point-pattern with one centred cluster, no Benford conformity for any mix of point patterns,

Allocation of values to points–option 1: high values of the population were in random point-pattern, intermediate values in regular and random, while small values in clustered locations; option 2: big and intermediate values of the population were in random point-pattern, while clustered, regular and random locations had small values.

Results of the simulation are as follows:

*Scenario 1* - 1D high conformity, 2D high conformity–even if all 1D and 2D components always conform with Benford, the 3D distances behave differently: in case of values allocation as in option 1, they were Benford-like marginally acceptable in 60% and non-conforming in 40%, while in case of random allocation of values 3D were always non-conforming.

*Scenario 2* – 1D high conformity, 2D low conformity–this is the spatially weaker version of scenario 1—at any allocation of values, 3D distances were always non-conforming with Benford.

*Scenario 3* – 1D high conformity, 2D nonconformity—even if 1D components always conformed with Benford (50% close conformity and 50% acceptable conformity) and 2D distances were in 100% non-conforming, the 3D distances behaved extremely different: in allocation as in option 2, they were always Benford-like—in 100% close conformity, while in random allocation, they were never Benford-like–in 100% nonconformity.

*Scenario 4* – 1D nonconformity, 2D high conformity—even if 2D components always conformed with Benford (60% close conformity and 40% acceptable conformity) and 1D distances were always non-conforming, the 3D distances were always Benford-like–in allocation as in option 1: in 30% acceptable and in 70% marginally acceptable, in random allocation in 100% acceptable.

*Scenario 5*: 1D nonconformity, 2D nonconformity—both 1D and 2D components never conformed with Benford; the 3D distances differed in their behaviour–in values allocation as in option 2 they never were Benford-like, while in random allocation, they were always closely conforming.

As a classification of MAD statistic to Benford conformity classes may generate some bias, we have analysed the correlations between MAD numerical values for 1D, 2D, and 3D approaches (Fig 10). We clearly show that Benford MAD of 2D spatial distances is correlated with Benford MAD of 3D soc-geo distances strongly: *corr = 0.66* in case of non-random allocation of values to places and *corr = 0.51* in random allocation. The impact of population 1D distance on 3D soc-geo Benford conformity is smaller, or even negative. We conclude that the impact of spatial distribution is primary, while of population secondary. Allocation of population values matters for the 3D Benford conformity. In the simulation, it changed the correlations between 1D, 2D and 3D MADs. In the empirical analysis, we expect that allocation of values allows the whole system to be Benford-like, having given spatial and population distributions. We conclude that in a search for 3D Benford conformity, spatial 2D Benford-like mostly matters, while population 1D only slightly impacts.

## Urban location–from history to present

One can ask if there is any link between historical urbanisation paths and the above findings. The below presented historical overview proves that links are strong. Current location of cities is the cumulative output of centuries of human activity and results from different geographical and historical contexts. We are to show that the last 500 years of history explain the appearance of clustered, regular and random urban locations worldwide, while the XXth century was a time of evolution in population allocation. Each epoch, with its stage of development, political and economic systems, transportation modes and mobility patterns, food production and delivery systems, local and global interactions, created different settlement patterns [37]. Thus the history of urbanisation differs among continents.

The *regular pattern* emerges from archaeological studies for the Middle Ages in Europe [38] and Asia [39]. Location of cities as "nodal points" next to transport routes lowered the accessibility costs. The spatial separation of cities was often limited to one-day travel in a horse-drawn cart (25–30 km). Dense, uniform and well-organised settlement cities' networks

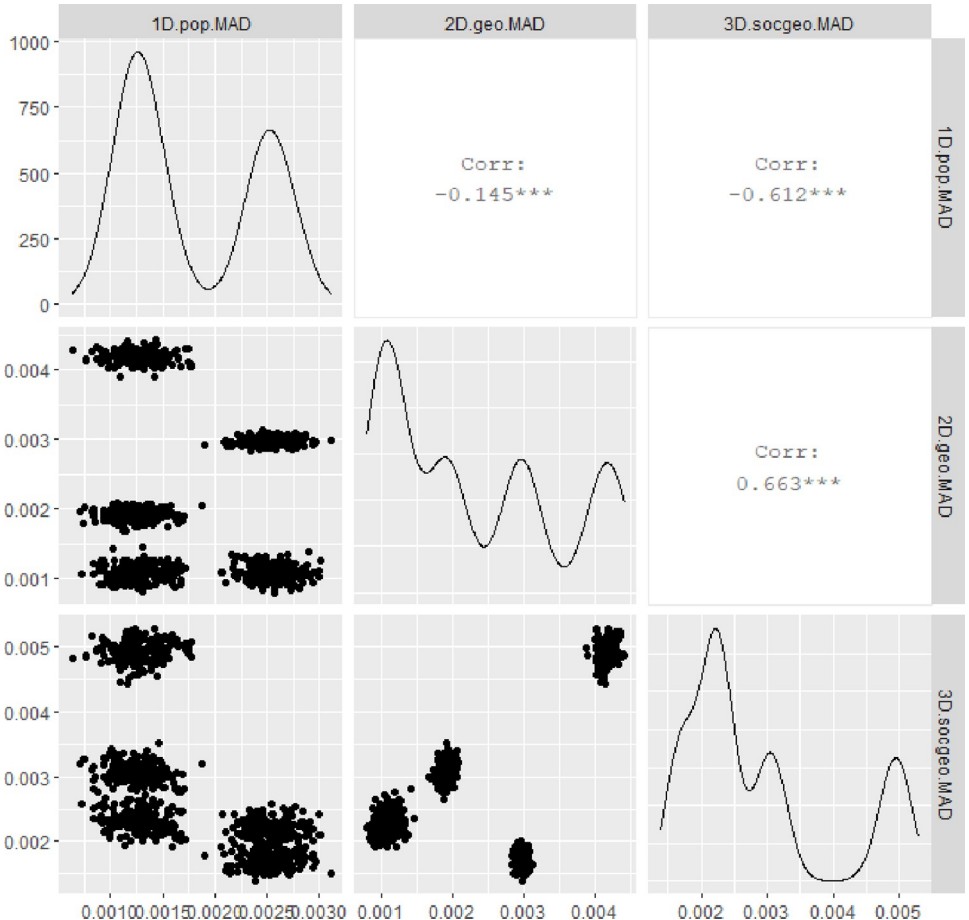

**Fig 10. Relations in simulated data.** Note: we analyse relations in simulated data between: a) MAD statistics for Benford conformity for 1D distances between population values (*MAD.pop.dist*), b) MAD statistics for Benford conformity for 2D distances between spatial points (*MAD.dist*), c) MAD statistics for Benford conformity for 3D distances between locations and population (*MAD.socgeo.dist*). Source: Own study with the use of GGally:: R package.

also emerged as cores of chiefdoms, similar in size and surrounded by forests and fields. Finally, as in XIX century's von Thünen model [40], the supply rule often made the locations next to forests, fields, rivers, and fertile valleys [41]. Those factors generated a relatively dense, uniform and well-organised settlement network, with local interactions only. With the significant role of political, administrative, religious and economic factors, the development of this pattern created the hierarchical urban systems that fit the regular in shape Christaller's Central Place Theory [42]. However, its range of impact and interactions is local only.

The *random pattern* is linked to natural conditions, and also magical and symbolic factors, which attracted and repelled cities location. In times of industrialisation, the deposits of coal and metals ores were attracting people to work. However, this first-nature geography is highly random in terms of spatial location compared to non-mining settlement patterns.

The *clustered pattern* exemplifies gateway cities. These port locations and forelands, which for the last 600 years mattered for international relations, connected the separated communities and markets. In McKenzie's gateway theory [43], ports, forelands, and cities established between two under-developed areas become central places and mainly were referred to North America [44] pioneering settlement. They grew on trade and got connected with regional settlement networks, often as centres of regular or irregular clusters.

XXth century, with strong economic development, significant technological and transportation progress, intensive trade interactions and high mobility, as well as strong political powers and worldwide urbanisation, made many cities expand. Core locations–playing the role of international and global cities, also caused the establishment of urban clusters around themselves, often with strong economic specialisation. Current world population maps [45] evidence high-density areas next to middle-size slow-path cities.

However, Africa and South America experienced other development paths. In Africa, except northern and Sub-Saharan cities with long ancient history, urban settlement started with XIX century–mostly at the coast as trading colonial stations, coastal forts, or cities for freed slaves, and continued until early XXth's [46]. The growth of those capitals boomed after gaining independence in the 1950s and 1960s. This intense urbanisation with a poor urban network made main African cities grow quickly [47]. Still, differently from European or US patterns [48]: less middle-size cities spread uniformly over the territory and fewer cities in general, much bigger primate cities at the cost. In South America, a highly urbanised continent, the cities were the legacy of the conquistadors' urbanisation [49] and British and USA colonial trade links [50, 51]. Urban sites were not randomly located, but for pragmatic reasons, as ports, or in inland next to rivers, fertile lands, with favourable winds. In contrary to other continents, many cities were abandoned or re-located [49]. Interestingly, urban geographers assessed South American cities' spatial location as far from natural but instead directed by the whim of the *plantadores de cidades* [52].

Today's urban locations have their roots in past centuries. Even if some old cities lost their urban status or even disappeared (as Peruvian Vilcabamba and Machu Picchu, Chinese Xanadu etc.), most ancient urban settlements are part of the current urban picture. This location pattern can be called "natural" as it was shaped over hundreds of years by natural forces of human existence. We observe that countries with long settlement history, and access to sea or ocean, experienced evolutionary mechanisms of establishing the urban sites, which results in Benford's law conformity. On the contrary, South American and African countries that generally have fewer cities or do not have an interior dense urban network or have no populous port cities due to a lack of ocean access do not exhibit Benford's law conformity.

## Conclusions

This study discovers new Benford's law properties, which opens the path for further research on complex systems. Firstly, it tests Benford distribution on distances in 1, 2 or 3-dimensional space, not on typical vectors of values, using urban population and location worldwide. 1D distance is between values of vector of population ($z$), 2D distance is a typical distance between points in geographical space ($xy$), and 3D distance is soc-geo distance referring to spatial separation and mass of $xyz$ point. Secondly, it shows how to simulate 1D, 2D and 3D systems which conform with Benford. Third, it refers statistical findings to urban settlement history.

In an empirical analysis of cities located worldwide, we show that 3D socio-spatial urban structure, where both location and population matter, most countries conforms with Benford. This confirms that human activity generates mechanisms that tend to natural distribution. One observes the evolutionary patterns. As cities were created and located over hundreds of years for different goals at diversified external and internal conditions (as mobility, population, functions, national borders etc.), their today's spatial structure does not conform with Benford. However, the population allocation adjusts the urban system to natural Benford-like distribution. This is a kind of long-term socio-geographic equilibrium, while nonconformity with Benford suggests the distortions in the past or presence. We show that the essential component of

natural Benford-like spatial pattern is the clustered point-pattern—modern urban geography confirms this trend as dominating the world urbanisation process.

In the statistical analysis, we discover how to construct a spatial point pattern that conforms with Benford. We call this *natural spatial distribution*. We prove that Benford-like mixture includes around 75%-80% of the non-centrally located clustered pattern and 10–15% each of organised and random distributions. We show that when simulating natural spatial processes, the often-used random point-pattern does not reflect reality, as nature is not random. In the simulation of the local population, we use triple *gamma* (1,1) distributions for specified centiles, which fit well empirical data, and for 1D distance conform with Benford. We also show that Benford's conformity of system needs mostly the natural or almost natural spatial distribution, while the distribution of values is of secondary importance.

We see the potential of usage our results in many areas–in those which till now could not confirm Benford-like patterns, as complex networks or astronomy, and in those where Benford was never applied, as spatial statistics and urban studies. Those results evidence that the geographic system of cities is a complex non-linear three-dimensional network, which is not random or independent. Benford conformity helps discovering hidden order which is the driver of observed spatial heterogeneity. As Benford law holds in natural datasets, we claim that this three-dimensional spatial pattern of urban settlement can be treated as natural, i.e. conforming with the ordinary course of nature.

## Author Contributions

**Conceptualization:** Katarzyna Kopczewska, Tomasz Kopczewski.

**Data curation:** Katarzyna Kopczewska.

**Formal analysis:** Katarzyna Kopczewska, Tomasz Kopczewski.

**Funding acquisition:** Katarzyna Kopczewska.

**Investigation:** Katarzyna Kopczewska, Tomasz Kopczewski.

**Methodology:** Katarzyna Kopczewska, Tomasz Kopczewski.

**Project administration:** Katarzyna Kopczewska.

**Resources:** Katarzyna Kopczewska.

**Software:** Katarzyna Kopczewska, Tomasz Kopczewski.

**Supervision:** Katarzyna Kopczewska.

**Validation:** Katarzyna Kopczewska, Tomasz Kopczewski.

**Visualization:** Katarzyna Kopczewska, Tomasz Kopczewski.

**Writing – original draft:** Katarzyna Kopczewska.

**Writing – review & editing:** Katarzyna Kopczewska, Tomasz Kopczewski.

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
