## [Decision Letter · Decision Letter 0]

8 Jul 2022

PONE-D-22-13819Natural spatial pattern – when mutual socio-geo distances between cities follow Benford's lawPLOS ONE

Dear Dr. Kopczewska,

Thank you for submitting your manuscript to PLOS ONE. After careful consideration, we feel that it has merit but does not fully meet PLOS ONE’s publication criteria as it currently stands. Therefore, we invite you to submit a revised version of the manuscript that addresses the points raised during the review process.

We look forward to receiving your revised manuscript.

Kind regards,

Randeep Singh

Academic Editor

PLOS ONE

Journal Requirements:

“No, The funders had no role in study design, data collection and analysis, decision to publish, or preparation of the manuscript.”

4. We note that Figures 1, 3, 4 and 7 in your submission contain map images which may be copyrighted. All PLOS content is published under the Creative Commons Attribution License (CC BY 4.0), which means that the manuscript, images, and Supporting Information files will be freely available online, and any third party is permitted to access, download, copy, distribute, and use these materials in any way, even commercially, with proper attribution. For these reasons, we cannot publish previously copyrighted maps or satellite images created using proprietary data, such as Google software (Google Maps, Street View, and Earth). For more information, see our copyright guidelines: http://journals.plos.org/plosone/s/licenses-and-copyright.

 a. You may seek permission from the original copyright holder of Figure 1, 3, 4 and 7 to publish the content specifically under the CC BY 4.0 license. 

5. We note you have included a table to which you do not refer in the text of your manuscript. Please ensure that you refer to Table 1 and 2 in your text; if accepted, production will need this reference to link the reader to the Table.

Reviewers' comments:

Reviewer's Responses to Questions

**Comments to the Author**

1. Is the manuscript technically sound, and do the data support the conclusions?

Reviewer #1: Partly

Reviewer #2: Yes

2. Has the statistical analysis been performed appropriately and rigorously? 

Reviewer #1: I Don't Know

Reviewer #2: Yes

3. Have the authors made all data underlying the findings in their manuscript fully available?

Reviewer #1: Yes

Reviewer #2: No

4. Is the manuscript presented in an intelligible fashion and written in standard English?

Reviewer #1: Yes

Reviewer #2: Yes

5. Review Comments to the Author

Reviewer #1: I tend to disagree with the authors that the use of “Benford distribution to analyse the geolocation of cities and their population in the majority of countries” is a pioneering work. Why?

Benford distribution is one of many heavy-tailed distributions such as Zipf’s law, Pareto principle (80/20 principle), Korcak's Law, Horton's laws, Gutenberg–Richter law, Bradford's law (1:n:n^2), Lotka’s law, and Moore’s law.

According to the previous study by B. Jiang and his co-workers, all countries’ cities follow Zipf’s law, with the exponent being around 1.0; see the following studies:

https://www.tandfonline.com/doi/abs/10.1080/13658816.2014.988715?journalCode=tgis20

Note that natural cities (or all human settlements that are objectively defined) can be defined not only at country level, but also at city level, or the city of the city level.

https://www.tandfonline.com/doi/full/10.1080/13658816.2018.1427754

It is not surprising to me at all that mutual socio-geo distances between cities follow Benford’s law, since Benford’s law is a much more relaxed law than Zipf’s law. If natural cities follow Zipf’s law, they will surely follow Benford’s law.

The authors are still constrained by the conventional thinking, i.e., population as city size. This conventional thinking is very backward since population is initially and essentially for census or administrative purposes rather than for scientific purposes. With the advance of geospatial big data, we should adopt new ways of thinking. The current state of the art on Zipf’s law is to use natural cities, with which all countries’ natural cities follow Zipf’s law without a single exception. Even for Singapore as a city country whose natural cities can be generated to fit Zipf’s law.

The essence of Benford’s law and Zipf’s law is the notion or recurring notion of far more smalls than larges. This notion has been formulated as the scaling law (https://link.springer.com/article/10.1007/s10708-014-9537-y), which can be manifested by head/tail breaks (https://en.wikipedia.org/wiki/Head/tail_breaks). I therefore suggest the authors use the state of the art concepts (natural cities, the scaling law, and ht-index) to re-examine their findings.

Reviewer #2: This paper applied Benford’s law in a spatial context to examine the geo-location of cities and their population. The authors justify the Benford-like spatial disputation of cities and inhabitants worldwide following the evolutionary process, which is a novel finding. The paper addressed a very important question which has not been comprehensively studied in previous literature. The English writing is acceptable but with minor grammatical errors. Overall, this paper is well-written and the structure of the manuscript is very clear. However, a few issues should be addressed before it can be published:

The authors mentioned a few studies on location and population of cities such as Dziecielski et al. (2021). However, how these studies approach the issue should be elaborated in detail. I suggest the authors provide more relevant studies to better position the contribution of this paper.

The structure of the paper should be revised. For example, the paragraph in line 153 talks about Zipf’s law, which fits better in a later section of result or discussion. The sentence “As our empirical results will show relatively poor Zipf's law conformity” should not be in the methods section. I suggest the authors reorganize the contents so that the flow is more smoothly and readers do not need to go back and forth.

The big versus small country of 100 cities is an interesting finding. However, as globalization puts cities of different countries in close relationships such as trading, migration and change of administrative boundaries, the border effect should be discussed to extend Benford’s law non/conformity.

The discussion in line 247 can be further expanded at the policy level as opposed to the formation of the natural spatial pattern.

Size or shape of the country. It seems the comparison provided on the four case studies is incomplete. What does the comparison imply? The four countries are at very different development stages, geopolitical locations, and with various economic/population sizes.

6. PLOS authors have the option to publish the peer review history of their article (what does this mean?). If published, this will include your full peer review and any attached files.

Reviewer #1: No

Reviewer #2: **Yes: **ChengHe Guan

---

## [Author Response · Author response to Decision Letter 0]

14 Aug 2022

Dear reviewers, 

thank you for your valuable comments. We have addressed all points you raised. All the changes were evidenced in track-mode version attached. Our answers were attached in separate file (with lines references to track-mode version). 

Best regards, 

Authors

---

## [Decision Letter · Decision Letter 1]

5 Oct 2022

PONE-D-22-13819R1Natural spatial pattern – when mutual socio-geo distances between cities follow Benford's lawPLOS ONE

Dear Dr. Katarzyna ,

Thank you for submitting your manuscript to PLOS ONE. After careful consideration, we feel that it has merit but does not fully meet PLOS ONE’s publication criteria as it currently stands. Therefore, we invite you to submit a revised version of the manuscript that addresses the points raised during the review process.

We look forward to receiving your revised manuscript.

Kind regards,

Randeep Singh

Academic Editor

PLOS ONE

Reviewers' comments:

Reviewer's Responses to Questions

**Comments to the Author**

1. If the authors have adequately addressed your comments raised in a previous round of review and you feel that this manuscript is now acceptable for publication, you may indicate that here to bypass the “Comments to the Author” section, enter your conflict of interest statement in the “Confidential to Editor” section, and submit your "Accept" recommendation.

Reviewer #1: (No Response)

2. Is the manuscript technically sound, and do the data support the conclusions?

Reviewer #1: (No Response)

3. Has the statistical analysis been performed appropriately and rigorously? 

Reviewer #1: (No Response)

4. Have the authors made all data underlying the findings in their manuscript fully available?

Reviewer #1: (No Response)

5. Is the manuscript presented in an intelligible fashion and written in standard English?

Reviewer #1: Yes

6. Review Comments to the Author

Reviewer #1: I have two questions for the authors to address:

“… However, those studies are usually based on socio-economic relations, are selective territorially, consider major cities only, refer to a relatively short time span and are not anchored in history”. This remark on “consider major cities only” is incorrect. In the study by Jiang and Ren (2019), all natural cities or all settlements were considered.

“As Benford law holds in natural dataset, we claim that spatial phenomena like urbanization, when created naturally, as Benford-like”. I agree on the remark, but unfortunately, government data about cities are unlikely to be natural, or cities are not natural. I am saying the definition of cities is not natural, since it excludes small settlements. Instead, human settlements (which are the concept of natural cities) are natural.

7. PLOS authors have the option to publish the peer review history of their article (what does this mean?). If published, this will include your full peer review and any attached files.

Reviewer #1: No

---

## [Author Response · Author response to Decision Letter 1]

6 Oct 2022

We have submitted the file with detailed answers and track-mode paper to highlight changes we have introduced.

---

## [Editor Report · Decision Letter 2]

7 Oct 2022

Natural spatial pattern – when mutual socio-geo distances between cities follow Benford's law

PONE-D-22-13819R2

Dear Dr. Kopczewska,

We’re pleased to inform you that your manuscript has been judged scientifically suitable for publication and will be formally accepted for publication once it meets all outstanding technical requirements.

Kind regards,

Randeep Singh

Academic Editor

PLOS ONE
---

## [Editor Report · Acceptance letter]

11 Oct 2022

PONE-D-22-13819R2 

Natural spatial pattern – when mutual socio-geo distances between cities follow Benford's law 

Dear Dr. Kopczewska:

I'm pleased to inform you that your manuscript has been deemed suitable for publication in PLOS ONE. Congratulations! Your manuscript is now with our production department. 

Kind regards, 

on behalf of

Dr. Randeep Singh 

Academic Editor

PLOS ONE